# Toward Efficient Inference for Mixture of Experts

**Haiyang Huang**[1]* **Newsha Ardalani**[2] **Anna Sun**[2] **Liu Ke**[3]*
**Hsien-Hsin S. Lee**[4]* **Shruti Bhosale**[2] **Carole-Jean Wu**[2] **Benjamin Lee**[5]*
[1]Duke University [2]FAIR at Meta [3]Washington University in St. Louis
[4]Intel Corporation [5] University of Pennsylvania
hyhuang@cs.duke.edu
{new, annaysun, shru, carolejeanwu}@meta.com
ke.l@wustl.edu
lee.sean@gmail.com
leebcc@seas.upenn.edu

## Abstract

Mixture-of-Experts (MoE) models have recently gained steam in achieving the
state-of-the-art performance in a wide range of tasks in computer vision and natural
language processing. They effectively expand the model capacity while incurring
a minimal increase in computation cost during training. However, deploying
such models for inference is difficult due to their large model size and complex
communication pattern. In this work, we provide a characterization of two MoE
workloads, namely Language Modeling (LM) and Machine Translation (MT)
and identify their sources of inefficiencies at deployment. We propose three
optimization techniques to mitigate sources of inefficiencies, namely (1) Dynamic
gating, (2) Expert Buffering, and (3) Expert load balancing. We show that dynamic
gating improves maximum throughput by $6.21$-$11.55\times$ for LM, $5.75$-$10.98\times$ for
MT Encoder and $2.58$-$5.71\times$ for MT Decoder. It also reduces memory usage by
up to $1.36\times$ for LM and up to $1.1\times$ for MT. We further propose Expert Buffering,
a new caching mechanism that only keeps hot, active experts in GPU memory
while buffering the rest in CPU memory. This reduces static memory allocation by
$1.47\times$. Finally, we propose a load balancing methodology that provides additional
robustness to the workload. Our code is available at `https://github.com/hyhuang00/moe_inference`.

## 1 Introduction

A machine learning model's predictive ability increases with the number of parameters. Model
capacity has grown at an exponential rate of $10\times$ per year [1], which in turn has driven demand for
computation. Mixture of Experts (MoEs) decouple model capacity from computational demands by
using conditionally, sparsely activated neural networks. They can reduce training costs yet improve
accuracy [2–4] for language modeling [5–8], machine translation [9], and image recognition [10, 11].

But training is only half the story. MoE inference is important yet challenging as large language
models are deployed for production services. Our experiments show MoE inference is relatively
inefficient, requiring much more time to perform the same number of calculations. In Section 3, we
show that MoEs are $15\times$ slower for language models and $3\times$ slower for machine translation compared
to their FLOP-equivalent dense counterparts. Distillation could shrink models and reduce latency but
harm model quality [6, 8, 2]. Optimizations could increase parallelism and GPU utilization, but they
narrowly target specific kernels for communication collectives and GPU computation [12, 13]. They

---

*Work done while interning/working/visiting FAIR at Meta.

38th Conference on Neural Information Processing Systems (NeurIPS 2024).

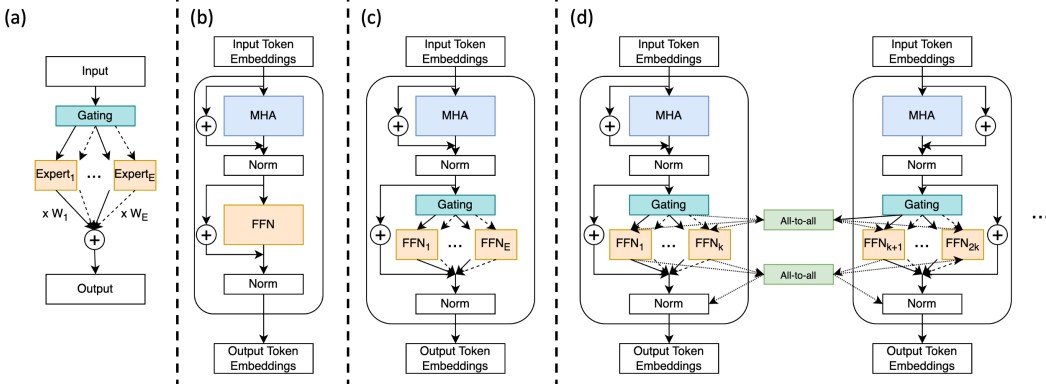

Figure 1: Illustration of MoE Models. **(a)** MoE module uses a gating function to assign inputs to experts [15]; **(b)** Dense Transformer Decoder Layer, which consists of Multi-head Attention (MHA) followed by a Feed-Forward Network (FFN); **(c)** MoE Transformer Layer in which an FFN block is replaced by a set of expert FFNs, which operate in parallel; **(d)** MoE Transformer with expert parallelism. Each device hosts a subset of experts. Tokens assigned to remote experts are dispatched via all-to-all communication.

lack a comprehensive analysis of inference costs and neglect inefficiencies in the MoE algorithms themselves.

*We explore optimizations for MoE inference to improve three important dimensions—token throughput, memory use, load balance—without degrading model quality.* We begin by identifying sources of inefficiency in MoE inference, breaking down latency and memory use across components of the model architecture. We find the gating function, which assigns tokens to experts, is a major contributor to MoE's high latency and large memory footprint.

We address MoE's inefficiency with **Dynamic Gating**, a new gating function that better matches each expert's computational capacity to its token assignments, thereby reducing communication and computation for placeholders. Dynamic gating reduces latency and memory use while enabling inference with larger batch sizes and fewer GPUs. We implement this new gating function on an open-source MoE Transformer [14]. Dynamic gating could be integrated with other optimizations for distillation, communication collectives, and GPU kernels for even greater benefit.

Furthermore, we develop optimizations that allow experts to better use GPU memory and cores. **Expert Buffering** exploits high temporal locality across experts. It allocates a fixed, but limited, amount of GPU memory for hot, active experts and relies on CPU memory to buffer all other experts. Less frequently accessed experts are brought into GPU memory as needed, significantly reducing demand for GPU memory. Expert buffering is orthogonal to existing memory management techniques such as offloading. **Load Balancing** mitigates severe load imbalance across experts. Although MoEs are trained with a loss function encouraging load balance, the token distribution during training often differs from that during inference. Our load balancing technique tracks and estimates expert loads (and their hosting GPUs) based on runtime expert activation data. It then redistributes tokens to balance the load, improving system robustness and reducing risks of out-of-memory errors and oversubscribed GPUs.

We implement and evaluate these optimizations for language modeling (LM) and machine translation (MT) tasks. These optimizations significantly improve inference throughput, memory use, and load balance. Moreover, they outperform the state-of-the-art and previously proposed optimizations.

## 2  Background

In this section, we aims to provide a basic background for readers unfamiliar with MoE transformers. A more detailed description on the MoEs and related works can be found in App. A.

Mixture-of-Experts (MoEs) use different models for different inputs to improve versatility and robustness [15]. An MoE consists of multiple, independent models (*i.e.*, experts) and a gating function that assigns inputs to experts. Each input activates only its assigned expert, allowing the model capacity to expand "outrageously" with more tractable increases in computational cost.

**MoE Transformer.** The Transformer model architecture has defined the state-of-the-art for computer vision and natural language processing [16, 17]. Illustrated in Figure 1, sparse MoE layers replace the FFN block in the Transformer architecture with an MoE block that consists of multiple expert FFNs. These layers use a gating function to decide which experts are most suitable for each token, and then routes tokens to their corresponding experts. Typically, a token is routed to one or two experts in a top-1 or top-2 gating policy. Compared to traditional Transformers, where the FLOP count per batch scales linearly with the number of parameters, MoE networks require much less computation and allow large models to be trained efficiently. MoE Transformers have reduced training costs for large models [2, 6–8] and achieved high accuracy in vision, text, speech and multi-task learning [18, 5, 19–21].

**Expert Parallelism.** MoE models present an interesting trade-off, requiring less computation but more memory usage than traditional Transformers of the same capacity. Expert layers deploy many additional FFNs, which increase model size and demands for GPU memory. GShard [2] addresses these challenges with expert parallelism, distributing workload across multiple GPUs. Each GPU holds a subset of expert FFNs and copy of all other parameters. All-to-all communication is required when distributing tokens to experts and collecting results from experts.

## 3 Mixture-of-Experts Characterization

We characterize MoE Transformers for Language Modeling (LM) and Machine Translation (MT) against FLOP-equivalent dense models. We study recent models on high-performance testbeds, and a detailed description can be found in Table 1 and 2 in App. B. The mini batch size is set to 8 for Language Modeling and 48 for Machine Translation, the largest feasible values under baseline.

### 3.1 Expert Activation

Under the baseline MoE design, an expert always processes a capacity of tokens regardless of the number of tokens actually assigned. Experts configured with excess capacity will suffer longer latencies and use more memory. But how much of this waste is incurred in practice? We answer this question by analyzing expert activations on several tasks.

**Language Modeling (LM).** We use three domains—Wikipedia, PubMed, Github, from the PILE dataset [22] as input following [8]. Fig. 5(a) in App. C.1 shows a highly imbalanced load across experts. Multiple hot experts consistently receive a large share of tokens, while other experts consistently receive a small share or no tokens at all. While all inputs show sparse expert activations, the set of hot experts and their hotness levels vary across domains.

**Machine Translation (MT).** We evaluate expert activation by performing translation from English to French, Japanese, and Austrian using validation data from NLLB-200 [9]. Fig. 5(b) in App. C.1 shows that MT also exhibits load imbalance and a small fraction of experts are hotter than others. Noticeably, decoder activation is extremely sparse, and differs from the encoder in one key aspect: it exhibit strong temporal locality. An expert may be active for several consecutive batch, then go inactive again. This temporal locality for hot experts is key motivation for our expert buffering optimization in Section 5.

### 3.2 Latency

Fig. 7 in App. C.2 examines inference latency. The MoE is $15\times$ slower for language models (LM) and $3\times$ slower for machine translation (MT). While MoE model will be more accurate than the FLOP-equivalent dense model, the MoE will exhibit much higher latencies than the dense. The finding that MoEs perform the same number of FLOPs but require much more time illustrates the need to mitigate the intrinsic inefficiency of MoE inference.

Fig. 9 in App. C.2 shows contributors to inference latency for different models and node counts. Prior studies attribute MoE's longer latency to frequent all-to-all communication [2], which is true for

MoE training where computation is distributed across many nodes and all-to-all communication is the main bottleneck. However, our characterization of MoE inference reveals that, while all-to-all communication occurs in multi-node deployments, it is less significant to latency than the computation for the gating function and experts. This finding informs our optimization in Sec. 4.

### 3.3 Memory Usage

Fig. 8 in App. C reveals contributors to MoE's significant demands for memory capacity. For LM, the dense model only requires 2.2GB on each GPU whereas the MoE requires up to 18.9GB, an increase of $8.6\times$. For MT, the dense and MoE models use 7.0GB and 21.2GB, respectively, an increase of $3.0\times$. Beyond the static model parameter usage, MoE models also use much more dynamic memory than dense models.

Fig. 10 in App. C illustrates dynamic memory use for the baseline MoE on the *Pear* cluster. A significant amount of memory is allocated during the gating and reordering computation, and then freed nearly instantaneously. Our close examination of the memory trace indicates that primary cause of this memory use is batch matrix multiplication within the static gating function. We address this challenge with dynamic gating in Section 4.

### 3.4 Load Imbalance

The root cause of MoE's high latency and memory use lies in its static gating policy. Recent implementations assume experts' computational loads are balanced [23, 12, 5, 8]. Under this assumption, token distribution can be simplified and implemented with all-to-all collectives, which we detail later in Fig. 2a(1). Inefficiencies arise when assumptions about load balance fail. If the gating function assigns fewer tokens than an expert's capacity, the remaining capacity is filled with placeholders (*i.e.*, zeros). If more tokens are assigned than the expert's capacity, excess tokens are dropped, with their information retained only by residual connections. Dropping tokens harms accuracy, so capacity $C$ usually set to large values to prevent information loss and accuracy fluctuations. However, large capacities increase latency and memory use. We find that it can leads to a waste as large as $12.8\times$ for LM and $64\times$ for MT (see App. C.3 for detail).

We study whether waste during inference is avoidable. If the workload is well balanced and token allocations across experts are comparable, we could reduce waste by simply scaling down expert capacity. On the other hand, if expert activation is sparse, scaling down capacity risks dropping tokens and harming model accuracy.

## 4 Dynamic Gating Optimization

Our characterization reveals a gap between assumptions and practice regarding load balance across experts, which gap leads to poor performance and resource inefficiency during inference. Although prior studies also notice load imbalance across experts [13, 12], they retain a gating policy that increases expert capacity in response to severe imbalance. Such a policy seeks model accuracy by ensuring the most heavily loaded experts do not drop tokens. But such a policy also exacerbates the high latency and inefficient memory use we have observed.

We propose dynamic gating that tunes efficient, variable capacities for experts. This improves upon static gating, which inefficiently sets large, fixed capacities. Changing the gating policy to permit dynamism is non-trivial. Major MoE implementations rely on statically set expert capacities to ensure all messages sent with all-to-all collectives are equally sized [13, 8, 10]. The NCCL all-to-all primitive requires recipients to pre-allocate memory for messages between GPUs. With equally sized messages, each GPU knows the required memory size beforehand. However, when message sizes vary due to dynamic gating and differing token assignments, a lightweight message is needed to notify each GPU recipient of its incoming tensor size.

**Static Baseline** Figure 2a(1) illustrates static gating, which constitutes our baseline. Here, $S$ represents the sequence length, $C$ represents the Capacity Factor, $E$ represents the number of total experts, and $D$ represents the dimension of each token. The gating function generates expert assignments and translates them into $E$ dispatch masks, each with dimension $(S, S \times C)$. Entries in the mask are generated as follows. If token $i$ is assigned to expert $e$ and the $e$-th mask still has

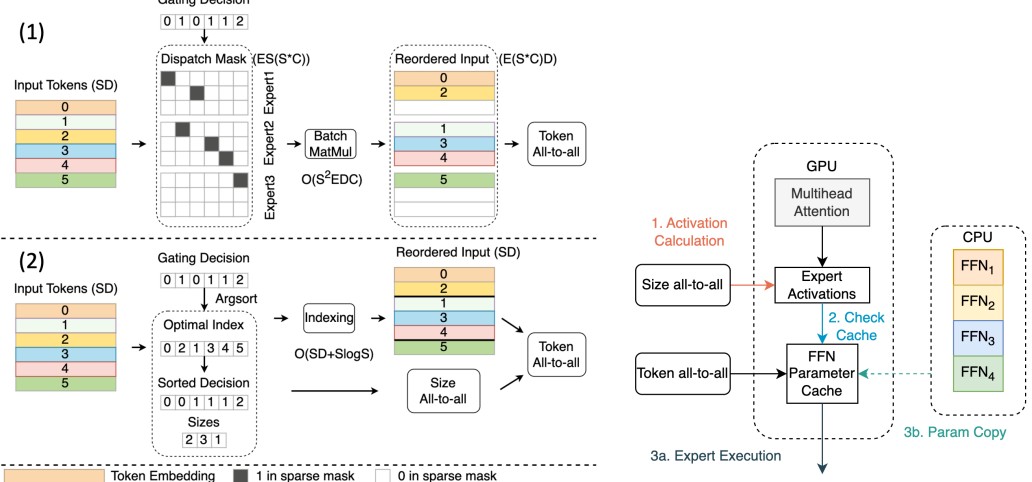

(a) Comparison for gating implementation.

(b) Illustration of Expert Buffering.

Figure 2: (a) Comparison between the static gating in [2, 8] and our implementation of dynamic gating. We assume E=3, S=6, C=0.5 and top-1 gating in this example. (See Sec. 4.)(b) Illustration of the Expert Buffering mechanism. We move the expert parameters to CPU memory to reduce burden on GPU memory. (See Sec. 5.)

capacity, the $i$-th column of the first empty row is marked 1 and other entries are marked 0. This process produces a highly sparse mask, which is a tensor with dimension $(E, S, S \times C)$ that contains at most $S$ 1's. Input tokens are multiplied with the mask to reorder inputs into $E$ sets of inputs, each with $S \times C$ tokens, indicating the assignment of tokens to experts.

**Dynamic Capacity for Gating** Figure 2a(2) illustrates our new dynamic gating procedure, which transfers a variable number of tokens to experts and devices. The procedure simplifies token distribution by transforming a vector of gating decisions. First, the procedure performs an argsort to generate indices that sort the decision vector by expert ID. Second, it uses the indices to produce a sorted decision vector. Finally, it counts the number of occurrences of each expert in the decision vector, thereby determining the number of tokens assigned to each expert.

Because the number of tokens assigned to each expert varies, dispatch requires two rounds of communication. First, experts are notified about the number of incoming tokens (*i.e.*, size) using an all-to-all collective. This notification happens as soon as sizes are known, allowing its latency to be hidden behind other computation. In parallel with this first round of communication, input tokens are re-ordered with optimized indices and then split based on sizes. Second, the gating function transfers the actual tokens with another all-to-all.

After all experts process their assigned tokens, tokens are collected with another all-to-all and sent to their original devices. Tokens are restored to their original order. This re-ordering is typically implemented using batch matrix multiplication but, as in the dispatch stage, the multiplication could be replaced with a more efficient indexing operation.

**Costs and Benefits** Dynamic gating complexity is $O(SD + S \log S)$ where $S$ is sequence length and $D$ is token dimension. The dispatch requires a sort of $O(S \log S)$, a bin-count of $O(S)$, and an indexing operation of $O(SD)$. The complexity of dynamic gating is much smaller than that of static gating, which requires batch matrix multiplication of $O(S^2 EDC)$ to reorder tokens such that those assigned to the same expert are contiguous. Dynamic gating eliminates the dispatch mask and avoids the multiplication. Instead, it uses an indexing operation that directly places tokens in the desired order.

Beyond this complexity analysis, dynamic gating incurs only modest additional communication costs. An additional all-to-all notifies experts about the number of incoming tokens, a single integer that is communicated at very low cost, which only 20 $\mu$s on average in our experiments.

In exchange for these modest costs, dynamic gating offers several significant benefits. It enhances model robustness by ensuring tokens are not dropped. When an expert receives more tokens, it adjusts its capacity accordingly, preventing load from exceeding capacity and thus avoiding token loss.

Additionally, dynamic gating improves computational efficiency by eliminating empty placeholders. When fewer tokens are sent, the expert is notified of the small load, avoiding unnecessary memory allocation and communication bandwidth associated with placeholders.

## 5  Expert Buffering Optimization

Our analysis of expert activation patterns indicates that, although some experts are often inactive, all experts are activated at least a few times across time and batches. This observation motivates our buffering mechanism, which judiciously offloads expert parameters to CPU memory, freeing GPU memory to hold frequently activated experts and enable larger batch sizes.

Figure 2b illustrates the buffering mechanism. During inference, each GPU hosts a number of experts and receives their corresponding tokens. An active expert is one that receives tokens for the current batch. If an active expert is not already present in GPU memory, a memory copy transfers expert parameters from CPU memory into GPU memory. The transfer of parameters proceeds in parallel with the transfer of tokens, allowing the buffering mechanism to overlap data movement and hide latency.

Scarce GPU memory is managed with an eviction policy that accounts for expert activation patterns. First, the policy evicts the experts that are not active in the current batch as these experts are less likely to be used in the near future (*i.e.*, concept of temporal locality). Second, the policy evicts experts with the Last In, First Out (LIFO) policy.

The choice of the LIFO policy is rooted in how recent MoEs have been implemented. When a single GPU hosts multiple experts, the MoE executes experts sequentially by increasing order of their IDs. Suppose the GPU hosts $E = 4$ experts and caches parameters for 2 experts. Further suppose that the batch activates experts 1, 2 and 3. The MoE Transformer first fetches and executes experts 1 and 2. It then evicts expert 2 and fetches expert 3. By evicting expert 2 instead of expert 1, the cache retains the expert with the shortest re-use distance.

No prior work exploits unique MoE characteristics to optimize memory use during inference. As a caching strategy tailored for MoEs, expert buffering is orthogonal to prior memory efficiency schemes and can be seamlessly integrated with other techniques, such as offloading [24, 3], for even greater memory savings.

## 6  Load Balancing Optimization

Our analysis of token distribution indicates severe computational load imbalance across experts. GPUs that host hot experts can become oversubscribed and vulnerable to out-of-memory errors. GPUs that host cold experts can idle while waiting for others to complete their computation. These observations motivate our load balancing technique, which colocates heavily loaded experts with lightly loaded ones using run-time activation data.

Let $P_{mn}$ denote expert placement where $m \in \{1, \ldots, E\}$ is expert ID and $n \in \{1, \ldots, D\}$ is device ID. When $P_{mn} = 1$, the $m$-th expert is placed on the $n$-th device. Let $A_{mb}$ denote expert activation where $m$ is expert ID and $b \in \{1, \ldots, B\}$ is batch ID. Each value in $A_{mb}$ is the fraction of tokens from batch $b$ assigned to expert $m$.

The placement problem can be reduced to the multi-way, number partitioning problem [25], which is NP-hard. Moreover, this optimization should be constrained such that each GPU hosts the same number of experts. This constraint balances memory use across GPUs and simplifies communication processes. The problem can be formulated as follows.

$$\min \max_{m,b} \left| \sum_n P_{mn} A_{mb} - \frac{1}{D} \right| \text{ subject to } \sum_m P_{mn} = \frac{E}{D} \ \forall n$$

**Greedy Balancing.** We implement a greedy algorithm to optimize the assignment of experts to GPUs. The algorithm sorts experts by their average historical load $\tilde{A}_m$ and sequentially assigns experts to

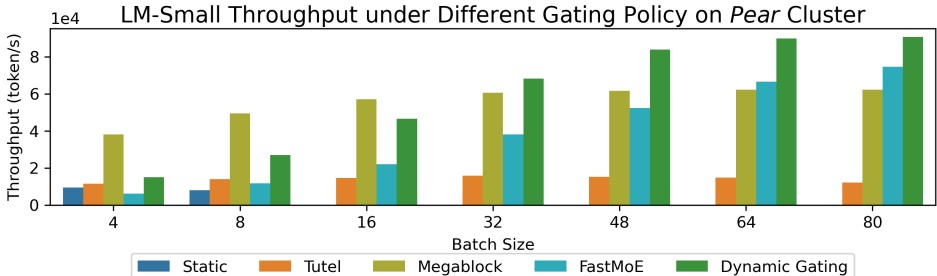

Figure 3: Throughput Comparison (*Pear*). Dynamic gating outperforms static, Tutel, and FastMoE consistently. It outperforms Megablock as batchsize scales. Missing bars represent infeasible combinations of policy and batch size.

GPUs in descending order. Each step in the sequence assigns an expert to the GPU with the smallest expected load $\sum_m P_{mn}\tilde{A}_m$. A GPU is excluded from consideration once it reaches designated load.

**Anti-Correlation Balancing.** Although the Greedy algorithm is effective when expert activations are independent (LM, MT-Encoder), it is less effective when activations are correlated (MT-Decoder). With correlated experts, the number of activations that are estimated from historical data $A_{mb}$ is a poor indicator for load. We address this challenge with Anti-Correlation Balancing. Let $S_{ab}$ denote the Pearson correlation between experts $a$ and $b$ that is observed in historical data. We revise the estimate of a GPU's expected load to $\sum_m P_{mn}(\tilde{A}_m + 0.5 * S_{am})$. This revision reduces load and tends to improve expert buffering performance for MT-decoder.

# 7 Evaluation

There are few direct baselines for efficient MoE serving. We obtain robust baselines by re-implementing and adapting previously proposed optimizations for inference. We preserve output quality and deliberately exclude token dropping, which harms quality [26].

**Methods.** *Fairseq* is our baseline MoE implementation [8]. *Tutel* improves latency and memory use with custom kernels [13]. It replaces the gating function's sparse matrix multiplication with a hash table lookup to distribute tokens, and creates a custom cumulative summation kernel to reorder inputs. *FasterMoE* organizes experts into fine-grained groups, combining token communication and expert execution within each group [27–29]. It overlaps the communication for one group with execution for another group. FasterMoE also places hot experts on each GPU, reducing communication. *Megablock* uses a custom kernel to execute experts and accelerate operations on a block-sparse matrix, which organizes token inputs and outputs [30, 29]. Nevertheless, the custom kernel depends on features in advanced GPUs, and may compromise backward compatibility. For Megablock, we adapt the experts' MLPs to omit the bias term aligning with Megablock's structural constraints.

**Clusters.** Table 2 details our experimental clusters. We use *Apple* to characterize MoE workloads (Table 1) and study the impact of our proposed optimizations. Due to limited machine availability on the *Apple* cluster, we perform additional experiments on *Pear* with NVIDIA's Ampere. Although our Ampere GPUs provide advanced features for Megablock's custom kernels, they offer limited memory capacity and restrict our experiments to a single node and smaller LM workloads. We report an average of multiple throughput experiments.

## 7.1 Impact of Dynamic Gating

**Single-Node.** Fig. 3–4 indicate dynamic gating significantly increases throughput for varied batch sizes, tasks, and clusters. LM throughput increases by $6.21\times$ and $3.32\times$ when compared against Fairseq and Tutel, respectively. Similarly, MT-encoder's throughput increases by $5.75\times$ and $5.33\times$ while MT-decoder's increases by $2.58\times$ and $1.88\times$. Beyond throughput increases, dynamic gating permits larger batch sizes by replacing the large dispatch mask with assignment indices and sorted decision vectors.

Compared to other methods, our dynamic gating technique outperforms FasterMoE by up to $2.49\times$, given the same batch size, by avoiding GPU kernel launch overheads. Moreover, our dynamic gating

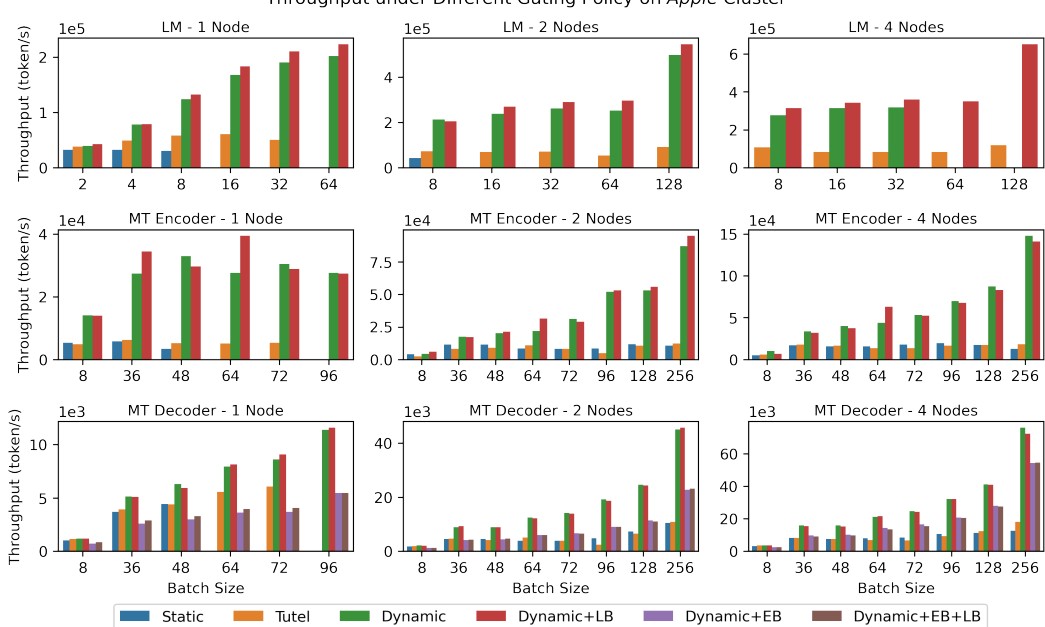

Figure 4: Throughput Comparison (*Apple*). Dynamic gating reduces memory use and communication, enabling larger batches and faster processing compared to static gating. Expert buffering (EB) trades latency for reduced memory use while maintaining high throughput. Load balancing (LB) can further improve latency. Note LB is relevant only for dynamic gating where each expert receives a different number of tokens. Missing bars indicate infeasible combinations of policy and batch size.

technique scales to larger batch sizes (and thus higher throughputs) better than Megablocks. Although dynamic gating underperforms for small batch sizes (4–16), it outperforms by increasingly large margins as batch size scales (32–80). When batch size is 80, dynamic gating outperforms Megablock by $1.46\times$, demonstrating better scalability. We argue that performance under large batch sizes is more important, as small batch sizes are uncommon in real-world inference workloads, particularly in online serving scenarios. Inference with small batch sizes limits throughput due to the reduced computational intensity. With the help of online batching [31], larger batch sizes can be easily achieved and become more frequent than the small batch size case. We also provide a detailed analysis on the underlying reason for the behavior in App. D.1.

**Multi-Node.** Fig. 4 shows performance when the system scales to two and four nodes. Dynamic gating tunes expert capacity to token load, thereby eliminating wasted communication for placeholders. More efficient communication translates into throughput gains when MoE models are deployed across multiple nodes. Dynamic gating improves throughput by up to $11.55\times$, $10.98\times$, $5.71\times$ for LM, MT-Encoder, and MT-decoder, respectively, when compared against Fairseq's static gating.

**Memory Use.** Fig. 11 assesses the impact on gating policy on memory use. Dynamic gating reduces memory use by eliminating the mask for token dispatch and by avoiding computation for placeholders. On the *Apple* cluster, memory use for activations falls by 79.6%, from 6.29GB to 1.28GB, when performing LM inference with a batch size of 8. Similarly, memory use falls by 44.2%, from 1.89GB to 1.05GB, when performing MT inference with a batch size of 8.

When activations use memory more efficiently, the MoE system can support larger batch sizes and achieve higher performance. On the *Apple* cluster, dynamic gating permits LM and MT to use batch sizes of 64 and 96, respectively. These batch sizes are $8\times$ and $2\times$ than those permitted by Fairseq's static baseline.

## 7.2 Impact of Expert Buffering

**Cache Miss Rate.** Each GPU deploys a cache to hold a subset of the experts assigned to it. When an expert is not found in GPU memory, the system incurs additional latencies to transfer the desired expert from CPU memory to GPU memory. For *Apple*'s MT tasks, we vary cache size from 16

experts—which accommodates all experts, fully occupies GPU memory, and offers zero reduction for static memory use—to 1 expert, which offers a 32% reduction in static memory use.

For each cache size, we report the global worst-case cache miss rate. The miss rate is global because it counts misses based on accesses across all GPUs and their memories. The miss rate is worst-case because it reports the highest miss rate across all GPUs and batches. This metric conservatively estimates the least efficient scenario in cache utilization, offering valuable insights for predicting throughput reduction.

Fig. 12(a) indicates cache miss rates for our LIFO policy approximates those from Belady's MIN, the theoretically optimal policy. The figure also indicates that cache miss rates improve most when cache capacity is greater than 5 experts per GPU or 40 experts across 8 GPUs. This observation corresponds with our prior finding that, on average, over 90 experts are not assigned any tokens by the MT-decoder.

**Throughput.** Fig. 4 shows the impact of expert buffering for MT decoding. The cache accommodates 10 experts per GPU and 80 experts across 8 GPUs. This cache size is the point at which Fig. 12(a) indicates diminishing benefits for cache misses.

For single-node inference, buffering negates some of the throughput gains from dynamic gating as cache misses impact performance. Nonetheless, dynamic gating and expert buffering together are still competitive with our baselines. For multi-node inference, dynamic gating and expert buffering together report throughput gains of $2.21\times$ and $4.30\times$ over baselines for two and four nodes, respectively.

**Memory Use.** Latency rises as memory use falls. Fig. 11 indicates buffering experts on CPU memory reduces GPU static memory use, for expert parameters, by 2.25GB. But Fig. 13 reports that as cache sizes shrink and memory use falls, latency rises as more experts are transferred between CPU and GPU memories. As expert transfers consume the limited bandwidth between the CPU and GPU, data rates peak at 12 GB/s and increase latency. New technologies that enhance CPU-GPU bandwidth (e.g., NVIDIA's Grace Hopper) can mitigate these latency issues when GPU memory capacity is constrained and caching only a subset of experts is necessary.

### 7.3    Impact of Load Balancing

Load balancing (LB) can further improve latency. This optimization is particularly beneficial when applied to multi-node settings or combined with expert buffering because balance improves cache performance. Note LB is relevant only for dynamic gating where each expert receives a different number of tokens.

We analyze load with and without our balancing optimizations using activation data from Section 3.1. We separate this data into two halves: the first half of the activation data is used to generate a device assignment for each expert, and the second half to estimate load under generated assignments. We record Max load, which is the maximum share of the tokens received by a GPU across all batches. Max load is a worst-case scenario and assesses risks from out-of-memory errors. We also record Avg-Max load, which is the maximum share of the tokens received by a GPU averaged over all batches. Avg-Max estimates typical load and assesses performance risks from oversubscribed GPUs.

Fig. 14 indicates that Greedy balancing successfully equalizes expert load for LM, reducing loads per GPU from upwards of 0.6 to below 0.4. Fig. 4 shows how balanced load translates into performance. First, Greedy balancing increases throughput by up to $1.11\times$ and $1.19\times$ when compared against pure dynamic gating. Second, it permits larger batch sizes of 64 and 128 when LM is deployed on multiple nodes. Greedy balancing is similarly effective for MT-encoder.

Anti-Correlation balancing is robust to MT-decoder's correlated expert activations. It successfully reduces Max and Avg-Max load in most cases, but the balanced load produces only modest throughput gains of $1.02\times$ when compared against pure dynamic gating.

## 8    Conclusion

We analyze the behavior of standard MoE Transformer workloads, pinpointing their inefficiencies in inference latency and memory usage. We introduce a Dynamic Gating policy that significantly

enhances efficiency in terms of latency and memory demands during inference. Building on this, we propose an Expert Buffering mechanism, demonstrating its effectiveness in substantially reducing memory requirements for MoE inference deployment with a nominal increase in latency. Additionally, we implement load balancing, leveraging historical activation data and heuristic methods to bolster deployment robustness.

Recent years witness the boom of MoE LLM models in commercial applications, including API services and personal assistants [32–35], and as a result, much more research is needed for efficient inference of MoE models. Techniques such as heterogeneous experts and token-dropping [12, 34] could offer trade-offs between performance and quality, which in turn could be potentially supported by heterogeneous allocations of GPU resources. Heterogeneous experts would motivate even intelligent and dynamic gating functions that assign tokens based on task difficulty. More efficient interconnect (e.g., optical) could benefit expert movement as well as all-to-all communication, which currently remain a challenge.

## Acknowledgement

This work is supported by a National Science Foundation grant CCF-2326606 (Expedition in Computing) for the authors at Duke University and University of Pennsylvania. Any opinions, findings, conclusions, or recommendations expressed in this material are those of the author(s) and do not necessarily reflect the views of this sponsor.

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

# A    Related Work

While the MoE Transformer substantially reduces the training cost and FLOPS for large models, the outrageous size of MoE Transformers and the complex expert parallelism [2] poses obstacles for its deployment, including the high GPU memory requirement and the excessive communication overhead of expert assignment. Various approaches have been invented to relieve these obstacles. Switch Transformer [6] and ELSLM [8] use knowledge distillation to distill a large MoE Transformer into a dense model. While distillation reduces the number of parameters, only a small portion (about 30%) of the accuracy gain can be retained. The MoS strategy proposed in DeepSpeed-MoE [12] distills the knowledge to a smaller MoE Transformer with less layers and shared experts. SE-MoE [3] uses pruning to reduce the number of experts in the model.

WideNet [36] and MPoE [37] reduce the number of parameters by enforcing parameter sharing. Beyond reducing the parameters, other methods directly reduce computation and communication. The BASE Layer and Switch Transformer also reduce the number of experts each token is assigned to reduce the communication volume and computation. V-MoE [10] further reduces the number by dropping out a large portion of tokens. Hash Layer [5] replaces the gating layer with a precomputed hash function, which reduces the computation cost, but doesn't alleviate the communication overhead. As the MoE Transformer is a type of Transformer, techniques and optimized architectures that enhance Transformer inference speed may apply. Relevant examples include Reformer[38], Longformer[39], and Terraformer[40]. However, there is scant discussion of their application to MoE Transformers, and interested readers may find a detailed review in [41] and benchmarks in [42].

In addition to direct modification of model architecture and parameters, deployment strategies, such as offloading strategies and customized kernel functions are also being explored to reduce the GPU resource usage and latency for inference. Offloading and swapping strategies such as [43] swaps unused tensors form the GPU memory to the main memory to reduce the resource requirement. However, existing strategies can only be applied on dense models. Applying these strategies efficiently on conditional neural networks such as MoE is non-trivial, since the data flow graph cannot be constructed in advance due to the conditional computation. FastMoE [27, 28] dissects communication primitives and expert executions on group-basis to overlap these kernels, but it has not been tested on large number of experts. Tutel [13] and DeepSpeed-MoE [12] improve MoE model performance on datacenter-scale systems by combining system and architecture methods with tailored kernels for both Transformer and MoE layers, and specialized communication primitives. The approach combines expert parallelism, model parallelism, and tensor parallelism to significantly boost throughput and reduce latency. However, DeepSpeed-MoE is not designed to conserve GPU resources and therefore may be impractical for many academic users. Megablock [30] utilizes block-sparse matrix to organize token inputs and combines consecutive expert MLPs into a single kernel, but it requires advanced GPU architecture and does not support bias term in expert MLPs. SE-MoE [3] utilizes Ring Memory offloading to reduce GPU usage, achieving better throughput than DeepSpeed-MoE in low-resource scenarios. However, this approach does not leverage expert activation pattern from MoE Transformers. MAD-Max is a performance modeling framework that enables better computation and communication overlapping to improve training and inference throughput, but the work only focuses on traditional MoE-based recommendation systems [44].

Recent years witness the boom of MoE LLM models in commercial applications, including API services and personal assistants [32–35]. Among these models, two stand out for their notable advancements: the Mixtral series [32] and the DeepSeek-MoE series [34], both of which merit brief discussion.

The Mixtral series reduces the number of experts to eight, while replacing all dense transformer decoder layers with MoE layers. but replaced all dense transformer decoder layer with MoE layer. Such a design allows the model to be deployed on standard server workstations, which are typically equipped with eight GPUs. The small scale of the model ensures the model can be deployed without expert parallelism, placing it outside the scope of this paper.

On the other hand, the DeepSeek-MoE further extends the shared expert proposed in [12], and adopts fine-grained expert separation to mitigate the feature collapse of MoE model [45]. Our optimizations are relevant for DeepSeek-MoE because, even when the MoE layer activates multiple experts (e.g., six in 16B model), many of MoE's inefficiencies remain. Expert sparsity remains a problem because the total number of experts is large, especially due to DeepSeek-MoE's approach to fine-grained

| Task | Type | Size | E | M | C |
|------|------|------|------|------|------|
| *LM-Small* | Dense | 125M | – | – | – |
| | MoE | 15B | 512 | 2 | 0.05 |
| *LM* | Dense | 355M | – | – | – |
| | MoE | 52B | 512 | 2 | 0.05 |
| *MT* | Dense | 3.3B | – | – | – |
| | MoE | 54.5B | 128 | 4 | 1 |
| Task | Type | Layers | TD | HD | Vocab |
| *LM-Small* | Dense/MoE | 12 | 768 | 3072 | 51200 |
| *LM* | Dense/MoE | 24 | 1024 | 4096 | 51200 |
| *MT* | Dense/MoE | 48 | 2048 | 8192 | 256206 |

Table 1: Model detail for Language Modeling-Small (LM-Small), Language Modeling (LM) and Machine Translation (MT). Hyperparameters include number of experts (E), MoE layer interval (M), and expert capacity fraction (C). Model parameters include token dimension (TD), hidden dimension (HD), and vocabulary size.

| Cluster | *Apple* | *Pear* |
|---------|---------|--------|
| CPU | 2×Intel Xeon E5-2698 v4 with 700GB memory | 2×Intel Xeon Gold 5317 with 64GB memory |
| CPU-GPU | 16GB/s via PCIe 3.0 | 32GB/s via PCIe 4.0 |
| GPU | 8×NVIDIA Tesla V100, with 32GB memory connected by NVLink | 4×NVIDIA RTX A5000, with 24GB memory |
| # Nodes | Single Node and Multi Node | Single Node only |

Table 2: Experimental Clusters.

expert separation. In this setting, our optimizations for expert buffering and load balancing will reduce memory use and latency for multi-device inference. Using shared experts is tangential and our optimizations could be extended to support them. For example, expert buffering would lock any shared experts into the cache and prevent their eviction from GPU memory to CPU memory.

## B   Experimental Details

In this section, we provide additional details to the experiments we conducted in this paper. We use the Python native time module to record latency and, separately, PyTorch Profiler to collect detailed traces.

Table 1 details model parameters. The table specifies the number of experts ($E$), how often a FFN is replaced by an MoE ($M$), and each expert's capacity fraction ($C$). We use capacity settings recommended by [8, 9]. the table also details the model architecture and relevant parameters for language tasks.

## C   Workload Characterization Details and Additional Discussion

### C.1   Expert Activation

Fig. 5 and Fig. 6 illustrates the expert activation pattern measured over different workloads. In Fig. 5, each row representing a batch and each column representing an expert. Colors indicate expert load as a percentage of all tokens in the batch. Hot experts consistently receive a large share of tokens is marked by intense color lines, while other experts receive a small share is marked by lighter color lines.

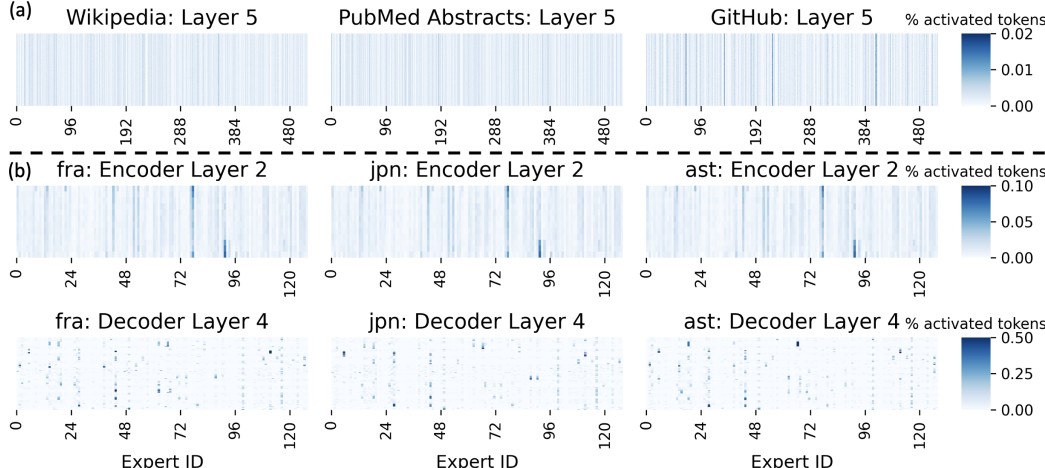

Figure 5: Expert activation patters on selected (a) LM and (b) MT layers. Activation reported as a percentage of all tokens in batch. Activations exhibit significant, consistent imbalance on all tasks.

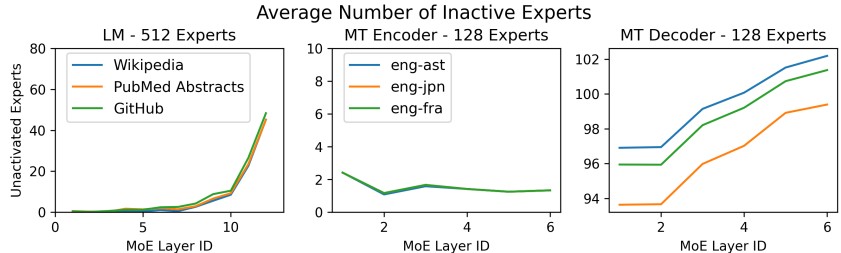

Figure 6: Average number of inactive experts for (a) LM, (b) MT encoder, (c) MT decoder. LM and MT encoder activate most, if not all, experts. MT decoder exhibits extremely sparse activations.

In Fig. 5(b), we can see that certain experts in both the encoder and decoder receive a large share of all tokens, nearly half of those in the full batch. Many experts receive few tokens. Encoder activation is mostly dense as most experts are almost always activated whereas decoder activation is extremely sparse.

Encoder activation patterns are similar across tasks. The encoder captures source language properties (English), which are the same across the three translation tasks. Surprisingly, decoder activation patterns are also similar across tasks despite differences in decoder architectures and target languages (French, Japanese, Austrian).

## C.2 Latency and Memory Breakdown

Fig. 7 shows the latency comparison between MoE and FLOP-equivalent dense model. For LM, the dense model requires 74.2ms whereas the MoE requires more than 1.09s. For MT, the dense model encodes and decodes in 101ms and 32ms, respectively, but the MoE requires 2.26s and 90ms.

**Effects of Model Scaling.** We optimize MoE inference based on our characterization of expert computation. Expert activation and locality depends primarily on the number of experts, a value that ranges from 128 to 512 in our studies. If the number of experts remains unchanged, the sparse activations that underpin our optimizations will persist. This is the most likely scenario as MoE models grow by increasing the size of each expert (*i.e.*, feed forward network) or the number of transformer layers rather than the number of experts.

Alternative scenarios seem less likely. If the number of experts were to shrink, activations may become less sparse, experts may exhibit greater locality, and our optimizations may be less helpful.

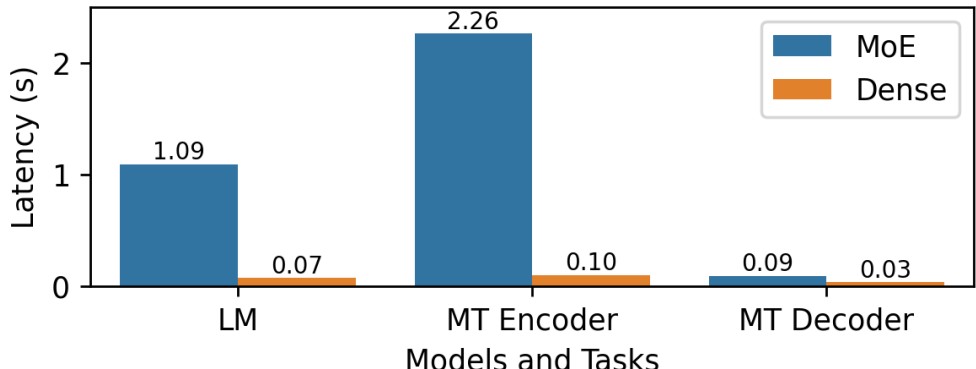

Figure 7: Inference latency for MoE and Dense models on single node. MoE is slower by 15× for LM, 22× for MT encoding, and 3× for MT decoding.

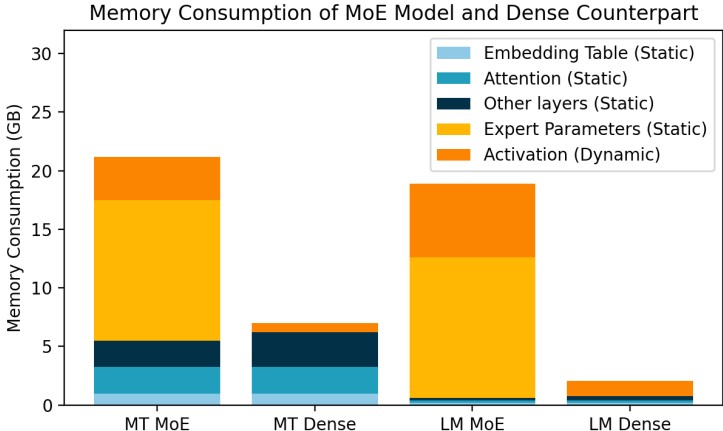

Figure 8: Inference memory use for MoE and dense models. MoEs use more memory for expanded model capacity (multiple experts) and model activations. Results shown using batch sizes of 8 and 48 for LM and MT, respectively.

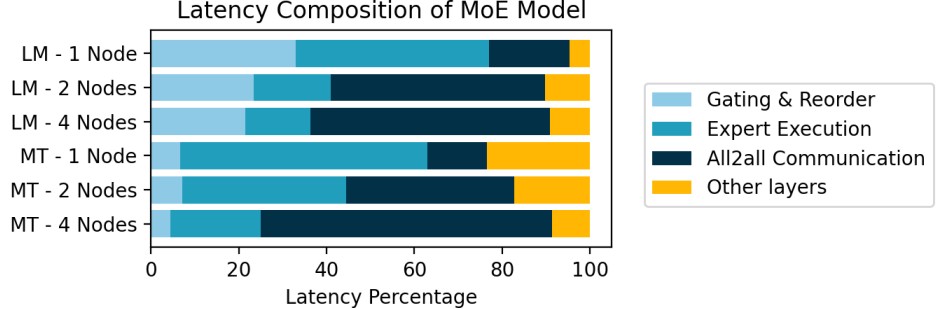

Figure 9: Inference latency for MoE models. Beyond communication, the gating function and expert execution are significant contributors to latency.

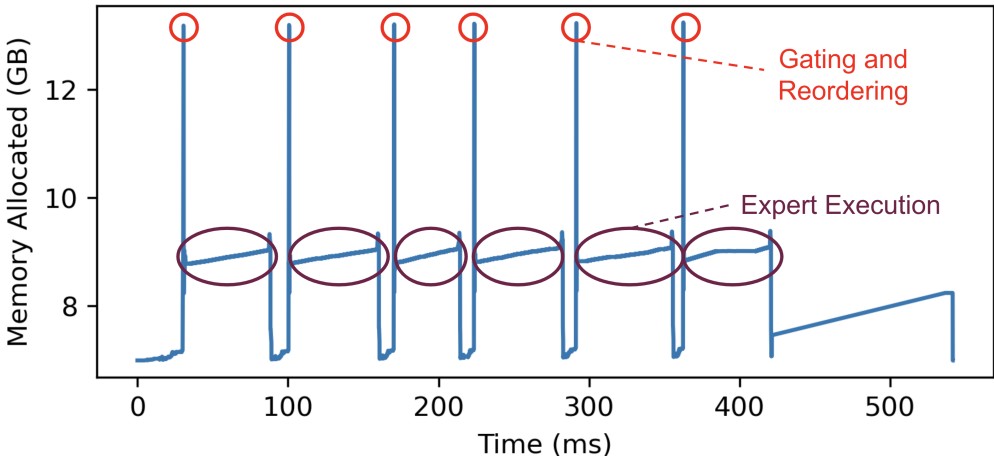

Figure 10: Memory trace of baseline MoE implementation on *Pear* cluster. Gating and reordering briefly allocates a large amount of memory.

However, reducing the number of experts would forgo the accuracy and efficiency benefits of scaling models with MoEs [8]. If the number of experts were to increase, activations may become more sparse and experts may exhibit less locality. This trend would make our optimizations more helpful but seems unlikely as more experts would put further pressure on already scarce GPU memory capacity.

### C.3    Calculation on Waste Factor

Consider a sequence of $S$ tokens for Language Modeling. If the MoE specifies $E = 512$ experts and $C = 0.05$ capacity fraction, each expert is configured to process $ECS = 512 \times 0.05 \times S = 25.6S$ tokens. However, if the MoE implements top-2 gating and each token is processed by two experts, computation is required for only $2S$ tokens. Thus, experts are configured to compute on many more tokens than they actually receive. This waste factor for LM is $25.6S/2S = 12.8\times$.

Similarly, for Machine Translation, each expert is configured to process $ECS = 128 \times 1 \times S = 128S$ tokens, but the model requires computation for only $2S$ tokens due to top-2 gating. The waste factor is $128S/2S = 64\times$. Such waste indicates MoE models typically perform a large amount of excess computation and communication as well as consume a large amount of extra memory.

## D    Additional Experimental Results and Analysis

In this section, we present experimental results that are informative yet could not be included in the main text due to space constraints.

### D.1    Analysis on Latency

In this section, we provide a detailed analysis on the latency measured in Sec. 7.1. Specifically, we compare our method with FasterMoE as well as Megablock on the methodology.

FasterMoE implements separate communication and computation kernels, which are executed concurrently to overlap and hide their latencies. This feature of FasterMoE is important for training, which requires significant amounts of time in both kernels. But this feature also incurs significant kernel launch overheads, which are hard to justify for inference, which spends less time in communication.

The essential difference between dynamic gating and Megablocks is the difference between several dense matrix multiplications and a single sparse matrix multiplication. Dynamic gating assigns a varied number of tokens to experts and then performs several multiplications on dense matrices. This technique avoids computation on zeros and padding. And computation for dense matrices is more efficient than that for sparse, leading to higher FLOP rates and inference throughput.

In contrast, Megablocks concatenates experts into a large matrix and then performs a single multiplication on a blocked compressed sparse row (BCSR) matrix. The BCSR format also avoids computation on zeros and padding, but incurs overheads from matrix metadata (column indices and row offsets) and indexing multiple blocked sub-matrices. The inefficiency of sparse matrix computation grows with matrix size, which in turn grows with batch size. Although dynamic gating requires several dense matrix multiplications, the cost of launching these kernels depends only on the number of experts and is constant as batch size increases.

### D.2 Memory Usage

Fig. 11 illustrates the dynamic memory usage of various methods on the *Pear*cluster and the *Apple*cluster. Memory use for activations is indicated by brighter colors in the figure. Note that GPU memory usage is not solely attributable to expert parameters and activations. Additional factors, such as memory fragmentation and cuBLAS workspaces, may also contribute to memory pressure and out-of-memory errors. But because these elements are beyond the direct control of gating policy, our evaluation focuses primarily on the GPU memory allocated to the MoE model and model activations.

### D.3 Additional Figures and Results for Effect of Expert Buffering

Fig. 12 illustrates the cache miss rates under different policy. Figure 13 illustrates how the cache size affects the latency.

### D.4 Additional Figures and Results for Effect of Load Balancing

Fig. 14 illustrates the effect of load balancing on the maximum load. Results are normalized by total batch size, which means numbers represent the share of the total number of tokens each GPU will handle in the batch. We record Max load, which is the maximum share of the tokens received by a GPU across all batches. Max load is a worst-case scenario and assesses risks from out-of-memory errors. We also record Avg-Max load, which is the maximum share of the tokens received by a GPU averaged over all batches. Avg-Max estimates typical load and assesses performance risks from oversubscribed GPUs. Results show our policy successfully reduce risks from out-of-memory errors.

## E Other Potential Optimizations

We explored several other techniques beyond those already detailed in this paper. Although these explorations did not yield significant results, they may offer researchers insight or cautionary conclusions.

**Parallel Expert Execution.** Figure 9 indicates expert execution is a significant contributor to latency. Considering that each expert's MLP processes only a small fraction of tokens, we attempted to parallelize their execution by creating multiple CUDA streams (*e.g.*, 2 or 4) on each GPU and executing a distinct subset of experts with each stream. Separate CUDA streams parallelize the execution of expert kernels, potentially increasing GPU utilization and reducing latency. Although our experiments did indicate some overlap in expert execution, we found only modest performance benefits accompanied by an increase in memory use due to the additional CUDA streams. Although expert MLPs are relatively small, they are substantial enough to require explicit allocation of GPU resources.

**Expert Kernel Fusion.** In an effort to enhance data locality and minimize kernel launch overheads, we attempted to integrate separate expert MLPs into a unified kernel. We consolidated parameters of the first layer from all expert MLPs into a single tensor and consolidated those in the second layer into another tensor. We then implement a batch matrix multiplication kernel that concurrently executes the first and second layers across all experts. Our experiments did not indicate any performance benefits.

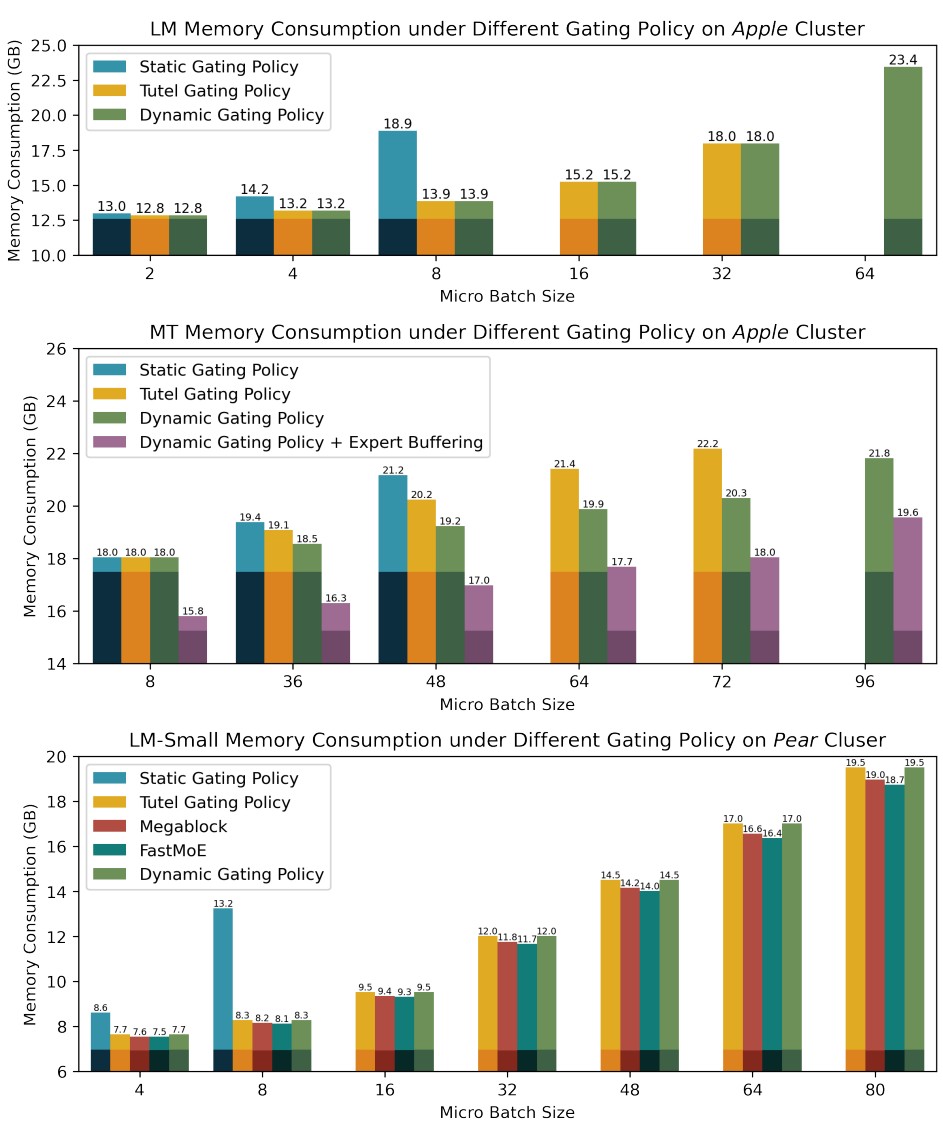

Figure 11: Memory Use Comparison (*Apple*, *Pear*). Dynamic gating reduces memory use, enabling larger batches. Expert buffering further reduces memory use for model parameters. Light shade is dynamic memory for activations. Dark shade is static memory for model parameters. Missing bars indicate infeasible cases.

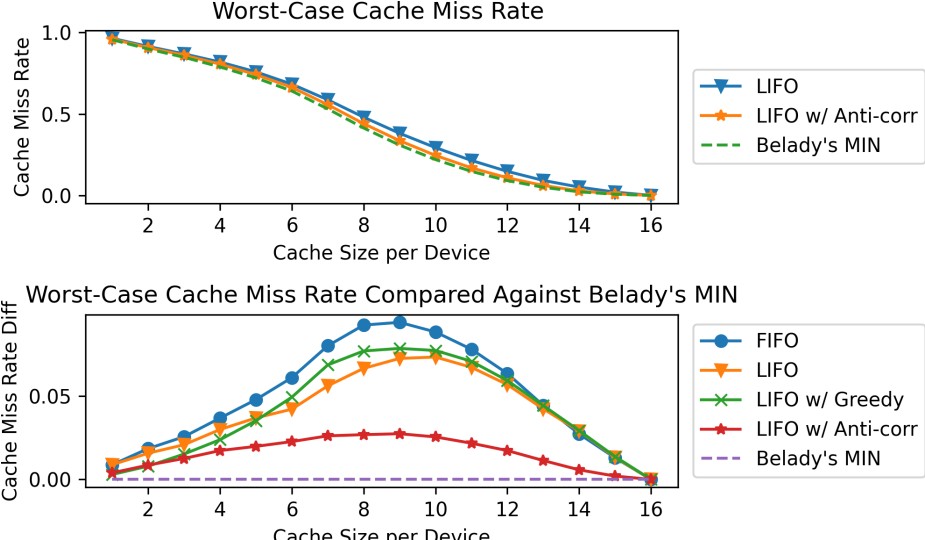

Figure 12: Cache Miss Rates. For a trace of MT-decoder's expert activations, **(a)** misses with and without load balancing and **(b)** misses compared against those from Belady's MIN. Miss rates are further reduced by load balancing.

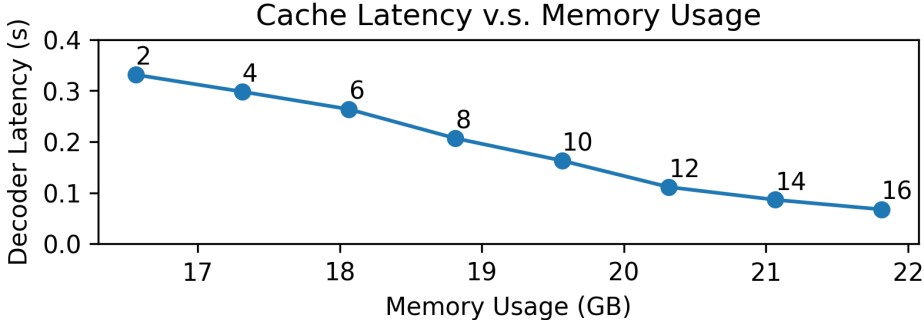

Figure 13: Cache Sizes vs Latency. For MT-decoder, decreasing the cache size decreases memory usage but increases latency. Cache size (reported in number of experts) per GPU marked on plot.

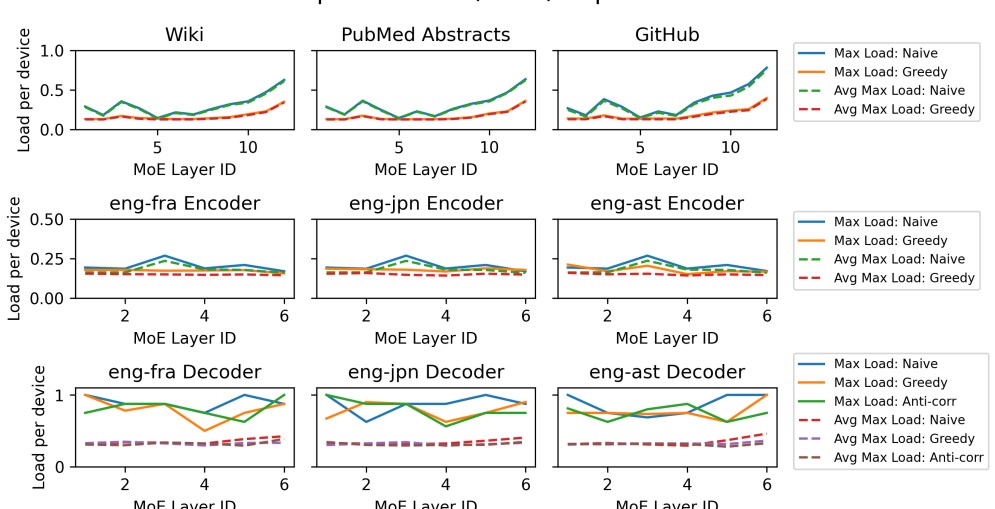

Figure 14: Load Balancing. Greedy and Anti-correlation algorithms balance load and reduce maximum load across GPUs, reducing risks from out-of-memory errors or poor performance from oversubscribed devices.

