# OpenReview forum: "Toward Efficient Inference for Mixture of Experts"
_NeurIPS.cc/2024/Conference — NeurIPS 2024 poster_

### Official Review · Reviewer_t1yX · 2024-07-09

**Soundness:** 2
**Presentation:** 2
**Contribution:** 2
**Rating:** 6
**Confidence:** 2

**Summary:**

The authors propose three techniques to speed up inference of mixtures of experts: 1) dynamic gating: during the all-to-all exchange process, the authors enable sending different number of tokens to each expert, which requires sorting and sending an extra message about the number of tokens; 2) expert buffering optimization: a last-in-first-out scheme to offload experts to CPU memory; and 3) load balancing optimization: greedily selecting experts with the lowest expected load.

**Strengths:**

1. The authors show consistent speed up over competing methods.
2. The authors show detailed latency and memory analysis in their appendix.

**Weaknesses:**

1. The techniques are not very substantial. The authors propose a collection of techniques that speed up inference, but the techniques are not thematic. As a result, I feel this paper may be much better suited for systems/industry tracks, where it would be significantly more relevant.
2. The authors do not provide performance analysis (everything is about throughput and latency). From my understanding (which may be wrong), dynamic gating and load balancing should affect performance, and I would be curious to know how it affects that.

**Questions:**

1. The authors show analysis for language modeling and machine translation. How about other tasks, e.g., summarization?

**Limitations:**

Limitations are sparsely discussed throughout the paper.

---

> ### Author Rebuttal · Authors · 2024-08-02
>
> We thank the reviewer for the review and suggestions. The reviewer raised several questions which we address in order:
> - **Is this paper suitable for NeurIPS?**
>
> We note that per the [call for papers](https://neurips.cc/Conferences/2024/CallForPapers), NeurIPS infrastructure track calls for submissions related to libraries, improved implementation and scalability, and distributed solutions, and we have submitted our paper to the infrastructure track. We believe our improved MoE implementation is a perfect match for this track.
>
> - **On performance impact.**
>
> Dynamic gating and load balancing will have no negative impact on the accuracy, perplexity, or BLEU score of the model. See common response 2 for a more detailed explanation.
>
> - **On summarization performance.**
>
> We note that summarization is increasingly performed by language models via zero-shot prompting (e.g. [1, 2]). Therefore, we believe experiments on summarization would be similar to those for language models.
>
> [1] Wu, Jeff, et al. "Recursively summarizing books with human feedback." arXiv preprint arXiv:2109.10862 (2021).
>
> [2] Adams, Griffin, et al. "From sparse to dense: GPT-4 summarization with chain of density prompting." arXiv preprint arXiv:2309.04269 (2023).

---

> > ### Comment · Reviewer_t1yX · 2024-08-14
> >
> > Thank you for the detailed response. After carefully reading the reviews and responses, I plan to increase my score to 6.

---

### Official Review · Reviewer_JXaG · 2024-07-13

**Soundness:** 3
**Presentation:** 2
**Contribution:** 2
**Rating:** 5
**Confidence:** 3

**Summary:**

This paper addresses the challenges of efficient inference for Mixture-of-Experts (MoE) models. The authors identify key inefficiencies in MoE inference, particularly in language modeling and machine translation tasks. They propose three main optimization techniques: dynamic gating, expert balancing, and expert load balancing. The authors implement and evaluate these optimizations, demonstrating improvements in inference throughput and memory usage compared to existing methods.

**Strengths:**

- The paper provides a detailed characterization of MoE workloads, identifying specific sources of inefficiency.
- The proposed techniques address key challenges in MoE inference.
- The optimizations show substantial gains in throughput.

**Weaknesses:**

- The paper lacks evaluation of existing representative MoE configurations, such as DeepSeek-MoE. Compared to traditional top-1 and top-2 gating, DeepSeek-MoE selects a larger number of experts, which could potentially impact inference performance. This omission limits the comprehensiveness of the study's comparative analysis.
- The proposed method demonstrates throughput advantages over MegaBlock only with large batch sizes. For smaller batch sizes, MegaBlock remains superior. This limitation somewhat restricts the applicability of the proposed method across different usage scenarios.

**Questions:**

- Is there a comprehensive experimental comparison available? Figure 3 in the main text only compares the throughput of different methods. However, the latency and memory usage of these methods are not reported. A more complete analysis including these metrics would provide a fuller picture of the proposed method's performance relative to existing approaches.

**Limitations:**

- The placement of most experimental figures in the appendix creates difficulties for readers. While this is understandable given the length constraints of conference submissions, it does impact the paper's readability. If additional space becomes available, the authors should consider including some of the key results in the main text. This would improve the flow of the paper and allow readers to more easily grasp the main findings without constantly referring to the appendix.

---

> ### Author Rebuttal · Authors · 2024-08-02
>
> We thank the reviewer for the very detailed review and suggestions. Here are our responses to each point raised by the reviewer:
>
> - **Including evaluation on representative configurations such as DeepSeek-MoE (2401.06066).**
>
> We appreciate the reviewer’s suggestion to include a discussion on the DeepSeek-MoE architecture. We will add citations and a detailed discussion in a later version of this paper. It is difficult to have a fair comparison with DeepSeek-MoE, because (1) the DeepSeek-MoE released implementation assumes single-device deployment and does not support expert-parallelism, and (2) the DeepSeek-MoE released implementation is based on Huggingface TGI framework, whereas our work is based on NVIDIA Megatron, and the custom kernels involved made it hard for us to perform an apple to apple comparison.
>
> We believe that our optimizations can also be applied to the DeepSeek-MoE architecture. Compared to traditional MoE Transformer architecture discussed in the paper, DeepSeek-MoE incorporates two additional strategies – (1) fine-grained expert separation and (2) the idea of shared experts. Below, we discuss the interaction of the two strategies and our optimizations.
>
> 1. Our optimizations are relevant for DeepSeek-MoE because, even when the MoE layer activates multiple experts (e.g., six in 16B model), many of MoE’s inefficiencies remain. Expert sparsity remains a problem because the total number of experts is large, especially due to DeepSeek-MoE’s approach to fine-grained expert separation. In this setting, our optimizations for expert buffering and load balancing will reduce memory use and latency for multi-device inference.
> 2. Using shared experts is tangential and our optimizations could be extended to support them. For example, expert buffering would lock any shared experts into the cache and prevent their eviction from GPU memory to CPU memory.
>
> - **On limitation of proposed optimization on small batch sizes compared to Megablock.**
>
> Please check the common response 1 for a detailed response and discussion.
> - **On detailed results for memory and latency.**
>
> Detailed memory usage for each model and optimization combination can be found in Figure 11, Sec. D.2 of the Appendix. In our experiments, the sequence length for each task was fixed, making throughput effectively inverse of mean latency, thus we omitted a separate figure for it. We will provide a detailed table of the measured mean latency in future revisions.
> - **On readability and figure placing.**
>
> We appreciate your understanding that the figures were moved to the appendix due to space constraints. We agree that the flow of the paper can be improved by relocating key figures back to the main text. We will prioritize moving more results to the main text as space permits in future versions.

---

> > ### Comment · Reviewer_JXaG · 2024-08-13
> >
> > Thank you for your detailed reply. I'll keep my original score.

---

### Official Review · Reviewer_dmgJ · 2024-07-13

**Soundness:** 4
**Presentation:** 3
**Contribution:** 4
**Rating:** 7
**Confidence:** 3

**Summary:**

This paper dives deep into the Mixture of Experts architecture, trying to identify its weaknesses and inefficiencies while coming up with novel solutions to improve the architecture in terms of token throughput, memory use and load balance. The authors find out that the static gating function that assigns tokens to each expert contributes most to the high latency and high memory footprint of MoE models. To this end, the authors come up with dynamic gating which improves the token throughput and reduces memory usage. Additionally the authors also add other optimizations such as expert buffering for better usgae of GPU cores and load balancing that takes into account the difference in token distribution at train and inference time. The authors demonstrate the efficacy of their method on Language Modeling and Machine Translation tasks.

**Strengths:**

1. I believe the paper is expertly written and provides excellent motivation for coming up with methods for making inference of MoE models efficient. I especially enjoyed reading Section 4 wherein each and every design choice is explained in detail.

2. The optimisations introduced in the paper are quite novel and lead to insane gains in throughput and memory usage. These optimisations are simple and can be easily used in practice for serving large MoE models. The results are pretty strong as well in terms of speedups gained.

**Weaknesses:**

1. While there are super detailed experiments on throughput and memory usage and I understand that those are the main talking points of the paper, I would have appreciated if there were some details about how the perplexity or BLEU scores were impacted by adding these optimisations. Some detailed insights into performance which we care about would have been great.

2. The authors have mentioned limitations of their work anywhere.

**Questions:**

Check weaknesses.

**Limitations:**

The authors have not a limitations section.

---

> ### Author Rebuttal · Authors · 2024-08-02
>
> We are encouraged to hear that the reviewer found our work to be thorough and our method effective. Here are our responses to each point raised by the reviewer:
> - **On details about how the perplexity or BLEU scores were impacted by adding these optimizations.**
>
> Our optimizations will not negatively impact the perplexity or BLEU scores of the proposed model. Please check our common response 2 for a detailed explanation.
>
> - **On creating a separate limitation section.**
>
> Due to the space limit, we choose not to create a limitation section, but we have mentioned limitations of our method in the main text as well as the Appendix. Specifically, limitations of our method include:
> - Dynamic gating may suffer from the kernel launching cost, resulting in slightly higher latencies compared to Megablock when batch size is small. We discuss this on line 273-277, and a more detailed discussion can be found in App. D.1.
> - Expert buffering trades latency for a smaller memory footprint. Additional CPU-GPU communication is necessary for offloading the memory. We mitigate the effect by designing a buffering strategy based on the LLM activation pattern and applying load balancing. Discussion can be found at line 311-313.
> - We discussed other failed approaches we tried in Appendix Section E.
>
> We hope that these responses help to clarify our points!

---

> > ### Comment · Reviewer_dmgJ · 2024-08-11
> >
> > Thank You for the clarifications. I would maintain my score of 7.

---

### Official Review · Reviewer_BGdS · 2024-07-23

**Soundness:** 4
**Presentation:** 3
**Contribution:** 3
**Rating:** 7
**Confidence:** 4

**Summary:**

This paper analyzed the behavior of standard MoE Transformer workloads and pointed out the bottleneck in inference latency and memory usage. Then it introduces a Dynamic Gating policy instead of static-size computation to improve the efficiency of the gating operation. It also proposes Expert Buffering which offloads inactive experts to CPU and Load Balancing which distributes experts and data in a more balanced way to cooperate with the Dynamic Gating policy to further improve the efficiency. Experiments on LM and MT demonstrate good performance of the proposed method.

**Strengths:**

1. The authors really do very detailed systematic and sound analysis of the bottlenecks in MoE and point out the modules that affect the efficiency, which is a strong motivation of the proposed method.
2. In view of the inefficiency of the gating operation, the authors proposed a dynamic gating policy, along with expert buffering and load balance to systematically improve the efficiency.
3. The experiments are convincing to address the motivation and support the methods.

**Weaknesses:**

1. In Section 4, line 155, the notations S, C, E, D lack explanation when they appear. Although I can guess out their meaning from the context surrounded, it really takes me time to understand these.
2. In Figure 3, it shows that when batch size is less than 32, Dynamic Gating performs worse than Megablock. During inference especially online service, I guess many requests are batchsize=1 and would be better to use Megablock?

**Questions:**

N/A

---

> ### Author Rebuttal · Authors · 2024-08-02
>
> Thank you for your enthusiastic and encouraging review of our work. Below are our responses towards each point and question raised in the review:
> - **The meaning of notation S, C, E and D in Section 4.**
>
> We appreciate the reviewer pointing out that these notations are not clearly defined when they appear. Here, S represents the sequence length, C represents the Capacity Factor, E represents the number of total experts, and D represents the dimension of each token. In future revision, we will add definitions to these notations in both Section 4 and Figure 2 to enhance clarity.
> - **Dynamic Gating performs worse than Megablock when the batch size is less than 32.**
>
> Please check our common response 1 for more details.

---

### Author Rebuttal · Authors · 2024-08-02

We would like to thank reviewers for providing us with valuable feedback.

We have taken note of the concerns raised by each reviewer and addressed them in detail. Here, we provide responses to the most shared questions first followed by a detailed response to each reviewer's concern in the rebuttal.

- **Dynamic Gating performs worse than Megablock when the batch size is less than 32.** (BGdS, JXaG):

While Dynamic Gating performs worse than Megablock when the batch size is less than 32, we argue that such cases are actually rare during inference, especially online serving. Performing inference with small batchsize will limit the throughput due to lower computational intensity of the workload, which is demonstrated in Fig. 4. Therefore, to improve resource utilization, request serving will be separated into multiple steps, and during each step, requests will be dynamically **re-organized into large batches** (i.e., Continuous Batching), as proposed by [1]. This technique has been adopted by recent LLM serving systems, such as vLLM [2] and TGI [3]. In a production system, multiple requests arrive at the node with short intervals between their arrival times, creating numerous opportunities for continuous batching. Therefore, we argue that the **performance on larger batch sizes is more important** for online serving, and our technique outperforms Megablock in such a scenario. We also note that a detailed discussion on the reason for the performance trend is provided in Appendix. D.1.

- **Impact of our optimizations towards the performance of the transformer model, in terms of perplexity, BLEU scores, or other metric of interest.** (dmgJ, JXaG, t1yX)

When expert capacity (the number of tokens accepted by an expert) is large, our optimized model is **mathematically equivalent** and as accurate as the baseline. When expert capacity is small, our optimized model may be more accurate (because it never drops tokens) than the baseline (which can drop tokens). Note that in baseline studies, evaluation is performed with large expert capacity and tokens are not dropped.


[1] Yu, Gyeong-In, et al. "Orca: A distributed serving system for Transformer-Based generative models." 16th USENIX Symposium on Operating Systems Design and Implementation (OSDI 22). 2022.

[2] Kwon, Woosuk, et al. "Efficient memory management for large language model serving with paged attention." Proceedings of the 29th Symposium on Operating Systems Principles. 2023.

[3] https://huggingface.co/docs/text-generation-inference/index

---

### Decision · Program_Chairs · 2024-09-25

**Decision:**

Accept (poster)

**Comment:**

Paper presents a timely and relevant investigation into optimizing the inference efficiency of MoE models, focusing on language modeling and machine translation tasks. The authors identify key bottlenecks and propose some optimization techniques to address these bottlenecks.

This paper presents valuable contributions to the field of efficient MoE inference, which is rapidly evolving and becoming prevalent both in research, developement but also in deployment (facing users). While some aspects require further refinement and clarification (see below), the work's strengths and potential impact warrant acceptance as a poster presentation.

- Limited Evaluation on Small Batches: One reviewer pointed out the performance drop of dynamic gating compared to Megablock with small batch sizes, which is a relevant concern for certain inference scenarios. While the authors addressed this by highlighting the prevalence of large batches in production settings, further discussion or potential mitigations for smaller batches would strengthen the paper.
- Limited Scope of Evaluation: While the focus on language modeling and machine translation is justified, exploring the impact of the proposed optimizations on other tasks like summarization (as suggested by a reviewer) would broaden the paper's applicability.
- Clarity and Presentation: Some reviewers noted the paper could benefit from improved clarity and presentation, especially regarding the placement of figures and tables.